# Trunk Control Measurement Scale (TCMS): Psychometric Properties of Cross-Cultural Adaptation and Validation of the Spanish Version

**DOI:** 10.3390/ijerph20065144

**Published:** 2023-03-15

**Authors:** Javier López-Ruiz, Cecilia Estrada-Barranco, Carlos Martín-Gómez, Rosa M. Egea-Gámez, Juan Antonio Valera-Calero, Patricia Martín-Casas, Ibai López-de-Uralde-Villanueva

**Affiliations:** 1Department of Physiotherapy, Faculty of Sport Sciences, Universidad Europea of Madrid, Villaviciosa de Odón, 28670 Madrid, Spain; javier.lopez3@universidadeuropea.es (J.L.-R.); cecilia.estrada@universidadeuropea.es (C.E.-B.); 2Doctoral Program in Healthcare, Faculty of Nursing, Physiotherapy and Podiatry. University Complutense of Madrid, 28040 Madrid, Spain; 3Department of Radiology, Rehabilitation and Physiotherapy, Faculty of Nursing, Physiotherapy and Podiatry, Universidad Complutense de Madrid, 28040 Madrid, Spain; juavaler@ucm.es (J.A.V.-C.); ibailope@ucm.es (I.L.-d.-U.-V.); 4Hospital Infantil Universitario Niño Jesús, 28009 Madrid, Spain; carlos.martin.gomez@salud.madrid.org; 5Spinal Unit, Department of Orthopedic Surgery and Traumatology, Hospital Infantil Universitario Niño Jesús, 28009 Madrid, Spain; 6InPhysio Research Group, Instituto de Investigación Sanitaria del Hospital Clínico San Carlos (IdISSC), 28040 Madrid, Spain

**Keywords:** cerebral palsy, trunk control, postural control, measurement scale, test-retest reliability, validity, psychometric properties

## Abstract

The aim of this study was to develop a Spanish Version of the Trunk Measurement Scale (TCMS-S) to analyze its validity and reliability and determine the Standard Error of Measurement (SEM) and Minimal Detectable Change (MDC) in children with Cerebral Palsy (CP). Participants were assessed twice 7–15 days apart with the TCMS-S and once with the Gross Motor Function Measurement-88 (GMFM-88), Pediatric Disability Inventory-Computer Adaptive Test (PEDI-CAT), Cerebral Palsy Quality of Life (CPQoL), and Gross Motor Classification System (GMFCS). Internal consistency was evaluated using Cronbach’s alpha, and the intraclass correlation (ICC) and kappa coefficients were used to investigate the agreement between the assessments. Finally, 96 participants with CP were included. The TCMS-S showed excellent internal consistency (Cronbach’s alpha = 0.95 [0.93 to 0.96]); was highly correlated with the GMFM-88 (rho = 0.816) and the “mobility” subscale of the PEDI-CAT (rho = 0.760); showed a moderate correlation with the “feeling about functioning” CPQoL subscale (rho = 0.576); and differentiated between the GMFCS levels. Excellent test–retest agreement was found for the total and subscale scores (ICC ≥ 0.94 [0.89 to 0.97). For the total TCMS-S score, an SEM of 1.86 and an MDC of 5.15 were found. The TCMS-S is a valid and reliable tool for assessing trunk control in children with CP.

## 1. Introduction

Cerebral Palsy (CP) is an umbrella term that describes a broad spectrum of developmental impairments in movement and posture that result in activity limitations. These impairments are attributed to non-progressive lesions that occur in the brain during the perinatal stage [1]. Children with CP often show motor disorders such as a lack of balance and postural control during functional tasks, which seems to underlie impaired trunk control [2,3]. Trunk control, in particular, seems to be related to the gait disturbances frequently found in children with CP [4,5]. As trunk control has been established as a key prerequisite for most activities such as postural changes and walking [6,7], deficits in trunk control could have an impact on the level of autonomy, the performance of daily activities, and even the quality of life in children with CP [8,9,10].

The latest clinical practice guidelines for the management of children and young people with CP recommend that the clinician should identify barriers that limit the acquisition of goals [6]. Assessing trunk control may help to identify specific features that make it difficult to perform functional abilities such as standing, sitting, reaching, or walking [4,7,8,9,10]; therefore, trunk control assessment has been highly recommended to determine intervention criteria and establish an adequate therapeutic program [4,11,12,13]. In addition, assessing trunk control seems to be the key to making decisions about technical aids and evaluating their effectiveness in structural and functional aspects [13,14].

Despite its relevance, there are few validated tools available to measure trunk control in children with CP [15,16,17]. The GMFM-88 is the most widely used scale for assessing motor function and postural control, as it has excellent psychometric properties [18,19]. However, the GMFM-88 does not specifically assess trunk control but rather motor development milestones [20]. The Trunk Impairment Scale (TIS) [21] was developed to assess trunk coordination and static and dynamic impaired trunk control in adults with stroke and Parkinson’s disease [21,22]. It has also demonstrated good reliability in the pediatric population with CP but may not be adequate to assess the multiple and complex clinical forms of CP found in the pediatric population [15,23]. In this framework, the original version of the Trunk Control Measurement Scale (TCMS) was developed based on the TIS but extended the items from it [24]. The TCMS assesses static and dynamic trunk control, including selective motor control and dynamic reaching. Furthermore, this scale provides information about the quality of the performance, observing whether there are compensations made during the execution of the items [24].

Following the development of the original TCMS, numerous studies have been conducted. On the one hand, some research has deepened our understanding of the relationship between trunk control and other dimensions of the International Classification of Function in children with CP. Indeed, the relationships between trunk control in children with CP and the level of motor function [25,26,27] and performance in activities such as walking are becoming apparent [4,5,7,13]. Trunk control has also been related to functions such as abdominal strength and incontinence [28]. On the other hand, several studies have shown the clinical relevance of the TCMS to compare the effects of different interventions [29,30,31,32,33].

The use of clinical assessment tools outside the countries and cultures where they were developed may lead to interpretation errors and differences in the meanings of the items. To minimize this risk and ensure that the assessment tools retain their psychometric properties, guidelines recommend a cross-cultural adaptation process [34,35]. The TCMS has been adapted to German [10,25], Korean [36,37], Turkish [23], and Tanzanian [27] languages andcultures with excellent results. Moreover, the use of the TCMS has been extended to other populations such as children with ataxic forms of CP [38], children with acquired brain injury and spinal cord injury [10], and even adults with spinal deformity [39,40].

Therefore, our aim was to develop the Spanish version of the TCMS (TCMS-S) and analyze its psychometric properties.

## 2. Materials and Methods

### 2.1. Study Design, Setting, and Participants

An observational study was carried out in accordance with the criteria established by the Consensus-based Standards for the Selection of Health Status Measurement Instruments guidelines (COSMIN) [41]. The study was approved by the ethics committee of the Bio-Medical Foundation of the Hospital Infantil Universitario Niño Jesús of Madrid (registration number: R-0066/20) and was conducted in accordance with the Declaration of Helsinki.

The sample was recruited from the in- or out-patient lists of the Hospital Infantil Universitario Niño Jesús of Madrid using a method of consecutive sampling by convenience. The selection criteria established were as follows: diagnosis of CP, between 5 and 19 years of age, able to sit without support, able to follow test instructions, and not having undergone any orthopedic surgery or botulinum toxin injections in the preceding 6 months [10,25]. The exclusion criteria were a diagnosis other than CP, unable to cooperate with or understand the test instructions, and unable to remain seated without the support of the trunk or feet [24,25]. Before the study, participants over 12 years signed the informed consent form and in the case of participants under 12 years of age, they gave their explicit verbal consent and their parents signed the form. Both participants and parents agreed to participate in the study.

### 2.2. Study Procedures

After confirming the inclusion criteria, all necessary sociodemographic and clinical data were obtained. Then, all face-to-face assessments were conducted on the same day, except for those included in the TCMS-S test–retest analysis, who were assessed on a second occasion between 7 and 15 days after the first assessment to minimize recall bias [24,26]. The same researcher administered the TCMS-S scale and the Gross Motor Function Measure-88 (GMFM-88). The total estimated time to complete the assessments was between 90 and 120 min. The TCMS-S assessments were also video recorded for later scoring to allow for the analysis of intra- and inter-observer reliability. The GMFM-88 was scored while the child was performing the tests [42,43,44], whereas the TCMS-S was scored later by analyzing the videos. To minimize rater bias as the same rater evaluated the GMFM-88 and TCMS-S, the rating of the videos was conducted two months prior to the face-to-face assessments [10].

All requirements for the protection of personal data were met. The processing, communication, and transfer of personal data of all patients complied with the provisions of Organic Law 03/2018 of 5 December on the Protection of Personal Data and the Guarantee of Digital Rights, as well as Regulation (EU) 2016/679 of the European Parliament and of the Council of 27 April 2016 on Data Protection (GDPR).

During the assessment, parents filled in the Pediatric Disability Inventory–Computer Adaptive Test (PEDI-CAT) and Cerebral Palsy Quality of Life (CPQoL) questionnaires accompanied by another researcher to resolve any doubts.

All data were recorded on an evaluation sheet and were checked before the end of the assessment while clarifying any questions by the participants or parents, in addition to communicating the results of the evaluations.

### 2.3. Outcomes

#### 2.3.1. Trunk Control Measurement Scale

The TCMS scale assesses seated trunk control in three dimensions. The maximum score is 58 points where 20 points correspond to static balance, 28 to selective movement control, and 10 to the ability to perform dynamic reaching. The items are scored from 0 to 3, with 0 being the inability to perform the task and 3 being the complete performance of the item. It is an active test where the evaluator gives verbal instructions, demonstrates the movement visually or by guiding the participant, and then asks the participant to perform the test. The best attempt out of three is scored. The administration time varies between 15 and 20 min [24].

#### 2.3.2. Gross Motor Function Measure-88

The GMFM-88 assesses gross motor function based on movements that are initiated by the child by collecting quantitative information (what the child does or does not do) but does not provide information on the quality of the movements. It is currently the gold standard for assessing gross motor function in pediatric neurological pathology and has been validated in Spain [19,45]. It consists of 88 items divided into 5 dimensions: (A) decubitus and turning; (B) sitting; (C) crawling and kneeling; (D) standing; and (E) walking, running, and jumping. Three attempts are allowed for each item and the best attempt out of the three is scored and the results are expressed as percentages. The administration time is variable but has been described as less than 60 min [20]. 

#### 2.3.3. Pediatric Disability Inventory Computer Adaptive Test

The PEDI-CAT measures a child’s ability and performance in activities of daily living. It contains a bank of 267 items for 4 independent domains: activities of daily living, mobility, social/cognitive, and responsibility [46]. It covers birth to 21 years and can be filled in by parents or professionals on a computer or tablet. The time to complete the questionnaire is approximately 13 min for each domain. [47].

#### 2.3.4. Cerebral Palsy Quality of Life

The CP-QoL questionnaire specifically assesses the quality of life of children with CP through questionnaires for parents and caregivers, as well as for children and adolescents. The questionnaire for parents of children aged 4–12 years consists of 66 items in seven domains: social well-being and acceptance, functioning, participation and physical health, emotional well-being, pain and the impact of disability, access to services, and family health [48]. The child self-report (children aged 9–12) contains 52 items with the same domains, except for access to services and family health. The questionnaire for parents of adolescents aged 13–18 years includes 72 items in seven domains: well-being and participation, communication and physical health, school well-being, social well-being, access to services, family health, and feelings about functioning. The teen self-report, as it appears in the child self-report, does not include questions about access to services or the caregiver’s health [49]. All questionnaires use a 9-point rating scale to measure how caregivers think their child feels (from 1 = very unhappy to 9 = very happy) and the scores are then converted to a scale from 0 to 100. Both versions have shown good psychometric properties [48,49,50].

### 2.4. Sample Size

The sample size calculation was estimated in accordance with previous studies [7,25] and was based on the criteria established in the COSMIN guide [41]. Thus, a sample size of 90 children with CP was determined to assess the validity of the content and a sample size of 30 children with CP was determined to assess the test–retest reliability. The sample size was considered very good and adequate for this population [25,41].

### 2.5. Linguistic Adaptation

To obtain the TCMS-S, a translation and back-translation process was carried out in accordance with international recommendations [34,35]. First, two translations from English to Spanish were carried out by two Spanish native speakers bilingual in English. Both versions were analyzed by a panel of experts composed of two translators and three physiotherapists with more than 10 years of experience in the field of CP and a synthesis version was obtained. Then, two back-translations into English were carried out by two English-speaking translators with a high level of Spanish and without knowledge of the original scale or the objective of the research, producing a final back-translated version. Lastly, an expert committee analyzed the scale and a pilot study of 15 patients with CP evaluated the clarity and feasibility of using the TCMS-S [34,35,51].

### 2.6. Data Analysis

The data were analyzed using SPSS v. 25 software (SPSS Inc., Chicago, IL, USA). The level of significance was set at *p* < 0.05. Shapiro–Wilk tests showed that most of the variables were not normally distributed.

#### 2.6.1. Validity

The internal consistency of the TCMS-S (total and subscales) was evaluated using Cronbach’s alpha, with a value > 0.7 being considered acceptable [52]. The construct validity was examined by analyzing the relationship between the total and subscale TCMS-S scores and the dimension scores and total scores of the GMFM-88, PEDI-CAT, and CP-QOL. The Spearman’s rho correlation coefficient was used to establish the correlations: few (rho < 0.30), low (rho = 0.30–0.50), moderate (rho = 0.51–0.70), high (rho = 0.71–0.90), and very high (rho > 0.90) [53]. The discriminant validity of the TCMS-S was assessed by comparing the total and subscale scores obtained across the different GMFCS levels. The group factor was analyzed using the Kruskal–Wallis test, and consecutive pairwise comparisons between adjacent GMFCS levels (I vs. II, II vs. III, III vs. IV) were performed using the Mann–Whitney U test with Bonferroni correction. The magnitude of the differences was calculated using *r* effect size: small (*r* < 0.3), medium (0.30–0.5), or large (>0.5) [54].

#### 2.6.2. Test–Retest Reliability

The test–retest reliability (7–15 days) [24,26] of the TCMS-S was assessed for each item separately, as well as for the subscales and total TCMS-S scores. For the score of each item, the percentage of agreement and kappa coefficients (kappa (dichotomous items) or weighted kappa (polytomous items)) were used. The kappa values were interpreted as follows: poor (<0.2), fair (0.21–0.40), moderate (0.41–0.60), substantial (0.61–0.80), and almost perfect (>0.80) [55]. For the absolute value of the total and subscale TCMS-S scores, the test–retest agreement was examined using the intraclass correlation coefficient (ICC): excellent (ICC ≥ 0.90), good (0.90 > ICC ≥ 0.70), fair (0.70 > ICC ≥ 0.40), and poor (ICC < 0.40) [56]. The standard error of measurement (SEM) and the minimal detectable change at a 95% confidence interval (MDC_95_) were calculated using the same criteria as a previous study [57]. In addition, the test–retest agreement was also examined using the method of Bland and Altman [58].

## 3. Results

### 3.1. Participants

Data were collected from a total of 105 participants; all were in- or out-patients of the Hospital Infantil Universitario Niño Jesús, recruited from the schedule of trauma and rehabilitation services. A total of nine patients were excluded for various reasons: one for not understanding the test instructions, one for a lack of cooperation, one for not being able to sit without support, and six for having a pathology other than cerebral palsy (one Steiner syndrome, one spinal cord injury, one medulloblastoma, one Charcot Marie Tooth, and two acquired brain damage).

A total of 96 participants (44 females, 52 males) with a mean age of 12.5 ± 3.3 years (range 6–19 years) were included in the validity study, and 35 of them were included in the test–retest analysis. Of the total sample, 47 participants (48.9%) were between 6 and 12 years and 49 participants (51%) were between 12 and 18 years. Of these, 37 had unilateral spastic CP, 32 had spastic diplegia, 3 had spastic triparesis, 18 had spastic tetraparesis, and 6 had ataxia. According to the GFMCS levels [59], 36 had a GMFCS level I with a mean age of 12.8 ± 3.3 years; 39 had a GMFCS level II with a mean age of 11.7 ± 3.6 years; 13 had a GMFCS III with a mean age of 12.8 ± 2.8 years; and 8 had a GMFCS level IV with a mean age of 13.6 ± 4.3 years.

### 3.2. Internal Consistency

The internal consistency of the TCMS-S total score was excellent (Cronbach’s alpha = 0.95 [0.93 to 0.96]). For the subscales of sitting balance, selective movement control, and dynamic reaching, the internal consistency was good–excellent, with a Cronbach’s alpha of 0.91 (0.88 to 0.93), 0.91 (0.89 to 0.94), and 0.85 (0.80 to 0.89), respectively.

### 3.3. Convergent Validity

The convergent validity of the TCMS-S was adequate, as the correlational analysis showed a statistically significant relationship between the TCMS-S total score and the GMFM-88 total score, as well as with children’s activities and quality of life (PEDI-CAT and CP-QOL, respectively; see Table 1). Specifically, the TCMS-S total score and its subscales showed their strongest correlations with the GMFM-88 (rho = 0.816), with high-magnitude correlations in all cases, except for the subscale “dynamic reaching” (rho = 0.624). In addition, the total TCMS-S score exhibited a moderate–high correlation (rho = 0.576−0.760) with the “daily activity” and “mobility” subscales of the PEDI-CAT (rho < 0.30), as well as with the “feeling about functioning” subscale of the CP-QoL. Although some were significant, the rest of the correlations, were small or negligible (rho ≤ 0.38).

### 3.4. Discriminant Validity

Concerning discriminant validity, the Kruskal–Wallis test showed statistically significant differences in the subscales and total TCMS-S scores (*p* < 0.001) for the GMFCS levels of the children with CP. In consecutive pairwise comparisons, statistically significant differences were observed between children classified as having a functional limitation level of I, II, and III on both the subscales and total TCMS-S scores (*p* < 0.001). In addition, in the total score, these differences were large for the comparison between children classified as level I versus level II according to the GMFCS (*r* = 0.66), as well as for the comparison between those classified as level II versus level III (*r* = 0.52). However, no statistically significant differences were found between children with a level III versus a level IV on the TCMS-S total score or on any of their subscales (*p* > 0.05). The multiple comparisons are presented in Table 2.

### 3.5. Test–Retest Reliability

The stability of the TCMS-S score obtained again at 10 days (test–retest) for each separate item was substantial to almost perfect (percentage agreement ≥ 77%; kappa ≥ 0.61 [0.36 to 0.86]), except for items “6a” and “9c left”, which showed moderate agreement (percentage agreement = 77–94%; kappa = 0.47–0.54; see Table 3).

With respect to the absolute agreement of the total and subscale scores of the TCMS-S, excellent agreement was found (ICC ≥ 0.94 [0.89 to 0.97]; see Table 4). The TCMS total score showed an SEM of 1.86 and an MDC_95_ of 5.15 (relative MDC_95_ = 12%). Interestingly, the “selective movement control” subscale had a slightly lower but excellent level of agreement compared to the other two subscales, both considering the score of each item separately and considering the absolute total score. In addition, the Bland– Altman plot for the test–retest agreement of the total TCMS-S score revealed no systematic bias (Figure 1).

## 4. Discussion

This study is the first to propose a Spanish cross-cultural adaptation of the TCMS, in addition to assessing its validity and reliability. In light of the results, the TCMS-S seems to have excellent psychometric properties.

### 4.1. Participants

The sample of our study is large and varied compared to other versions. A sample size similar to ours (96 participants) can be found in only two previous studies [7,25] and a wide age range of 6 to 19 years has only been included in two other studies [10,25]. Moreover, most studies only include participants with a diagnosis of spastic CP [7,23,24,27,36]. In our sample, as in the German and Turkish versions, we have also included ataxic forms of CP [10,25,26].

Although the TCMS was originally recommended for children of GMFCS levels I, II, and III [24], we included children of GMFCS level IV to evaluate the psychometric properties of the TCMS-S in this functional level as recommended by the authors of the original version [7,17] and other studies [8,10,25,27]. Children with GMFCS IV present difficulties with trunk control and posture, which is reflected by the lower TCMS scores, but this does not mean that it is unnecessary to identify where they need upper limb or other support to perform activities and improve their participation [6,7]. Given this, our results in this group should be interpreted with caution due to the small sample size from GMFCS IV.

### 4.2. Internal Consistency

The TCMS-S has shown excellent internal consistency for the total score and three subscales, supporting the construct validity of the scale. These results (about 0.90) were also noted for the original and Tanzanian versions [24,27]. No data on internal consistency were found in the other versions of the TCMS [10,25,26,36,37,38].

No ceiling effect was found because it did not exceed 15% in any case. Ceiling effect data were not described for the other versions, except for the Tanzanian version, which showed a ceiling effect in the dynamic reaching subscale [27].

### 4.3. Convergent Validity

The GMFM-88 was selected to analyze the convergent validity of the TCMS-S because trunk control is involved in most of the specific movements of the human body related to motor capacity [7,17]. The TCMS-S total and subscale scores had a direct significant relationship with the GMFM-88 total and subscale scores, which was consistent with the original and other versions [7,8,17,24,26]. Furthermore, the strongest relationship was found between the standing and walking dimensions of the GMFM-88 and the total scores of TCMS-S, demonstrating the importance of trunk control for standing and walking [4,5,7,13]. This could be the key to the planning of therapeutic goals related to mobility [6,11]. It may be of significance that in the present study, the same researcher scored both the GMFM-88 and the TCMS-S scales. The GMFM has demonstrated excellent intra- and inter-observer reliability when tested by evaluators with experience in the field and specific training in the use of the scale (ICC 0.99, 95% CI 0.99–1.00) [19,60,61]. Indeed, having a single evaluator can provide more reliable scores [10].

The TCMS-S demonstrated a good convergent validity with the PEDI-CAT, specifically with the PEDI-CAT mobility dimension. Similar results were recently found for the Turkish version of the TCMS [8]. Our results are also consistent with Marsico et al., who described the ability of the German version of the TCMS to differentiate between independent and dependent children in terms of mobility, as measured by the WeeFIM [10].

Moreover, our results indicated that there was a moderate relationship between the TCMS-S total score and PEDI-CAT activities of daily living. To the best of our knowledge, this relationship has not been studied in any version of the scale. It may be of clinical relevance to know how specific trunk control influences the performance of activities of daily living and self-care in order to develop specific intervention programs [62,63,64].

Regarding the correlation between the TCMS-S and quality of life, we observed a moderate correlation with the “functioning subscale”. This may indicate that children with greater trunk impairment scores have lower levels of functioning. This correlation was not observed for the other dimensions of the CPQoL, which could indicate that the quality of life in children with CP depends on other factors such as adaptations at home and school, social, and emotional support [48,50,65]. The ICF encouraged us to consider environmental factors and their role as facilitators or barriers in functional aspects and quality of life. This recommendation was reinforced by the results of the present study [66,67].

### 4.4. Discriminant Ability

As with the other versions of the scale, the TCMS-S was able to differentiate between all GFMCS levels, except between levels III and IV. For the original TCMS, Heyrman et al. [24] reported differences between all levels for the total and subscale scores of the TCMS, except between levels III and IV, specifically for the dynamic reaching subscale. Indeed, levels III and IV showed the greatest deficits in trunk control, presenting lower scores in the three subscales of the TCMS. However, these data should be interpreted with caution due to the small number (*n* = 8) of participants with GMFCS IV included in the present study. Moreover, overall, the TCMS-S had very good discriminant validity, similar to other versions of the scale [24,25,27].

### 4.5. Test–Retest Reliability

The test–retest reliability was analyzed for 35 children, following the studies of Heyrman et al. [4,24]. The ICCs of the total and subscale scores of the TCMS-S indicated excellent test–retest reliability. Similar results were found for the original version, as well as the other versions [10,24,26,27,36]. The test–retest analysis for each item showed high percentages of agreement (kappa coefficients), indicating good stability. However, item 6a reached only moderate agreement, as with the Korean version, whose authors highlighted the need to better define the goal of movement 32. Although the original TCMS [24] specified that item 6 involves leaning 45° forward with arms crossed in front of the chest, this movement could be challenging for some participants and this feeling of instability could be influenced by the surface of the table [25]. No data on item agreement analysis were found for any other versions of the TCMS [10,25,26,27,36,37].

The SEM and MDC values obtained for the TCMS-S were similar to those of the original, German, and Tanzanian versions [24,27], indicating a high level of agreement. However, other studies by Heyrman et al., Marsico et al., and Ravizzotti et al. reported slightly lower MDCs for the total TCMS score, ranging from 4.39 to 4.82 [24,25,27].

## 5. Clinical Implications

The TCMS-S scores had a moderate to high correlation with mobility and self-care in children and youth with CP. An in-depth analysis of the specific components of trunk control involved in specific tasks of daily living would be key to developing motor learning exercises that can transfer well to these tasks. It may, therefore, be interesting to add the TCMS to studies or interventions aimed at improving performance in daily life [13,68,69,70].

It is also necessary to consider the great variability in the clinical characteristics of children with CP. Within this framework, the results of the test–retest analysis of the TCMS-S are excellent, especially for the selective motor control subscale, which has been described as an exceptional indicator of motor control. This enhances the clinical value of the TCMS-S, as selective motor control is one of the main challenges faced by this population [4,7,24,71,72]. Furthermore, the TCMS-S has a small MDC, which supports its use in long-term and interventional studies to objectify relevant changes. A TCMS total score difference of more than 5 could indicate a true change in trunk control of the children/youth with CP [24,25,27].

## 6. Limitations

This study has several limitations. Firstly, the use of a single rater to administer both the GMFM-88 and TCMS-S scales may have introduced a rater bias, although this was minimized by strictly following the scale instructions. Moreover, using only one rater could lead to higher reliability and validity of the results compared to if the scales had been administered by two different raters [10,44]. Secondly, the scores were derived from video assessments. Although this procedure is widely described and accepted in scale validations and blinding of raters’ identities, it should be taken into account in the clinical use of the scale, e.g., to have one rater to administer the scale and another to give directions and score the items [10,42,44,73]. Furthermore, although the sample size was adequate for the main objective of analyzing the reliability and validity of the TCMS-S, it would be interesting to have a larger number of participants in each subgroup of age and functional levels to study the TCMS-S scores according to age and GMFCS level [7,17]. Specifically, it seems relevant that only eight participants were in level IV of the GMFCS so the results in this group of participants have been considered with caution.

## 7. Conclusions

The TCMS-S is a valid and reliable tool for assessing trunk control in children and youth with CP. It has shown significant internal consistency. The total and subscale scores of the TCMS-S exhibited strong convergent validity, as demonstrated by their strong correlation with the GMFM-88. Furthermore, the TCMS-S showed a moderate–high correlation with the “daily activity” and “mobility” subscales of the PEDI-CAT, as well as with the “feeling about functioning” subscale of the CP-QoL. The TCMS-S also exhibited adequate discriminant validity for the different functional levels of the GMFCS. These relationships may be useful for establishing therapeutic goals and re-evaluating treatment programs. In addition, the TCMS-S items exhibited excellent stability and the total and subscale scores of the TCMS-S showed excellent test–retest agreement. A difference of more than 5.15 points in the TCMS-S total score indicates a meaningful change in the trunk control of patients with CP.

## Figures and Tables

**Figure 1 ijerph-20-05144-f001:**
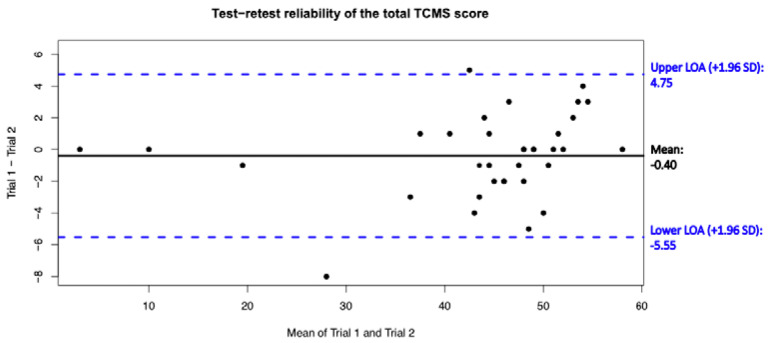
Bland-Altman plot for the rest-retest reliability of the Spanish version of the Trunk Control Measurement Scale total score. The mean difference is indicated by a solid horizontal line and the limits of agreement (95% Confidence Interval) are demarcated by the dashed horizontal lines. Dots stands for limits of agreement are at 95% CI.

**Table 1 ijerph-20-05144-t001:** Spearman correlations of the Spanish version of the Trunk Control Measurement Scale with gross motor function, self-care, and quality of life.

	Spanish Version of the Trunk Control Measurement Scale (TCMS-S)			
	Total	Static Sitting Balance	Selective Movement Control	Dynamic Reaching
**Gross Motor Function Measure-88 (GMFM)**				
Total score	0.816 **	0.738 **	0.795 **	0.624 **
Lying and rolling	0.575 **	0.537 **	0.544 **	0.577 **
Sitting	0.621 **	0.574 **	0.58 **	0.628 **
Crawling and kneeling	0.706 **	0.613 **	0.678 **	0.691 **
Standing	0.794 **	0.737 **	0.771 **	0.595 **
Walking, running, and jumping	0.790 **	0.711 **	0.777 **	0.603 **
**Pediatric Evaluation Disability Inventory-Computer Adaptive Test (PEDI-CAT)**				
Daily activities	0.585 **	0.428 **	0.601 **	0.474 **
Mobility	0.760 **	0.690 **	0.745 **	0.559 **
Social/Cognitive	0.380 **	0.239 *	0.422 **	0.144
**Cerebral Palsy-Quality of Life (CP-QOL)**				
Social well-being, acceptance, and participation	0.175	0.081	0.214 *	0.102
Feelings about functioning	0.576 **	0.519 **	0.580 **	0.406 **
Emotional well-being and self-esteem	0.268 **	0.255 *	0.267 **	0.077
Pain and impact of disability	0.034	0.036	0.022	0.025
School well-being	0.153	0.162	0.151	0.006
Access to services	0.235 *	0.203 *	0.239 *	0.219 *
Family health	0.260 *	0.237 *	0.256 *	0.224 *

* *p*-value < 0.05; ** *p*-value < 0.001.

**Table 2 ijerph-20-05144-t002:** Descriptive statistics and consecutive pairwise comparisons between children with different degrees of functional limitations according to the GMFCS for the Spanish version of the Trunk Control Measurement Scale.

TCMS-S	Mean ± SD; Median (IQR)				Between-Group Differences *p*-Value; *r* Effect Size (a) GMFCS I vs. GMFCS II (b) GMFCS II vs. GMFCS III (c) GMFCS III vs. GMFCS IV
	GMFCS I	GMFCS II	GMFCS III	GMFCS IV	
Total score	51.08 ± 4.67;	42.49 ± 5.79;	30.54 ± 10.14;	20 ± 12.56;	(a) *p* < 0.001; *r* = 0.66
	51 (48.75 to 55)	43 (38.5 to 47.5)	32 (24 to 38)	18 (14 to 24.5)	(b) *p* < 0.001; *r* = 0.52
					(c) *p* = 0.068; *r* = 0.41
Static sitting balance	18.97 ± 1.44;	17.56 ± 2.17;	13.08 ± 4.37;	9.57 ± 5.16;	(a) *p* < 0.001; *r* = 0.37
	20 (18 to 20)	18 (16 to 19.5)	14 (12 to 16)	9 (5.5 to 14)	(b) *p* < 0.001; *r* = 0.53
					(c) *p* = 0.110; *r* = 0.36
Selective movement control	22.39 ± 3.6;	15.56 ± 4.28;	10.54 ± 4.82;	5.71 ± 5.59;	(a) *p* < 0.001; *r* = 0.69
	22 (20 to 25.25)	16 (14 to 19)	11 (7 to 14)	5 (1.5 to 8)	(b) *p* = 0.002; *r* = 0.44
					(c) *p* = 0.057; *r* = 0.43
Dynamic reaching	9.72 ± 0.85;	9.36 ± 0.84;	6.92 ± 2.29;	4.71 ± 3.5;	(a) *p* = 0.006; *r* = 0.32
	10 (10 to 10)	10 (9 to 10)	7 (6 to 9)	4 (2.5 to 7)	(b) *p* < 0.001; *r* = 0.52
					(c) *p* = 0.140; *r* = 0.33

Abbreviatures: IQR, Interquartile Range; GMFCS, Gross Motor Function Classification System; SD, standard deviation; TCMS-S, the Spanish version of the Trunk Control Measurement Scale.

**Table 3 ijerph-20-05144-t003:** Agreement test–retest for each item of the Spanish version of the Trunk Control Measurement Scale.

Item	Kappa (κ)/ Weighted Kappa (κw)	Value (95% CI)	Agreement (%)
Static sitting balance			
Item 1 Item 2 Item 3 left Item 3 right Item 4 left Item 4 right Item 5 left Item 5 right	κ κ κ κ κw κw κw κw	1.00 (—) 0.81 (0.55 to 1.00) 1.00 (—) 0.78 (0.38 to 1.00) 0.81 (0.68 to 0.95) 0.90 (0.77 to 1.00) 0.75 (0.54 to 0.96) 0.91 (0.79 to 1.00)	100% 94% 100% 97% 83% 94% 86% 94%
Selective movement control			
Item 6a Item 6b Item 7a Item 7b Item 8a left Item 8a right Item 8b left Item 8b right Item 8c left Item 8c right Item 9a left Item 9a right Item 9b left Item 9b right Item 9c left Item 9c right Item 10a Item 10b Item 11a Item 11b Item 12a Item 12b	κ κ κ κ κ κ κ κ κ κ κ κ κw κw κ κ κw κ κw κ κw κ	0.47 (−0.15 to 1.00) 1.00 (—) 0.65 (0.03 to 1.00) 0.69 (0.44 to 0.94) 1.00 (—) 0.65 (0.03 to 1.00) 0.80 (0.58 to 1.00) 0.72 (0.47 to 0.97) 0.87 (0.705 to 1.00) 0.72 (0.47 to 0.97) 1.00 (—) 1.00 (—) 0.87 (0.73 to 1.00) 0.76 (0.58 to 0.95) 0.54 (0.26 to 0.82) 0.61 (0.36 to 0.86) 0.74 (0.54 to 0.94) 0.93 (0.80 to 1.00) 0.69 (0.44 to 0.94) 0.87 (0.63 to 1.00) 0.77 (0.61 to 0.92) 0.72 (0.42 to 1.00)	94% 100% 97% 86% 100% 97% 91% 89% 94% 89% 100% 100% 91% 83% 77% 80% 83% 97% 86% 97% 77% 91%
Dynamic Reaching			
Item 13	κw	0.88 (0.63 to 1.00)	97%
Item 14 left	κw	0.94 (0.81 to 1.00)	97%
Item 14 right	κw	0.95 (0.86 to 1.00)	97%
Item 15 left	κw	0.90 (0.69 to 1.00)	97%
Item 15 right	κw	1.00 (—)	100%

95% CI, 95% Confidence Interval.

**Table 4 ijerph-20-05144-t004:** Test–retest reliability of the Spanish version of the Trunk Control Measurement Scale total and subscales.

TCMS-S	Mean ± SD		ICC (95% CI)	SEM	MDC_95_ (Score); MDC_95_ (%)
	Trial 1	Trial 2			
Total score	43.51 ± 12.19	43.91 ± 11.65	0.98 (0.95 to 0.99)	1.86	5.15; 12%
Static sitting balance	8.89 ± 2.35	8.94 ± 2.33	0.99 (0.98 to 0.99)	0.24	0.66; 7%
Selective movement control	17.11 ± 4.54	17.37 ± 4.16	0.94 (0.89 to 0.97)	1.02	2.83; 16%
Dynamic reaching	17.51 ± 6.19	17.6 ± 6.05	0.95 (0.91 to 0.98)	1.35	3.75; 21%

Abbreviatures: ICC, Intraclass Correlation Coefficient; MDC, Minimal Detectable Change; SD, standard deviation; SEM, Standard Error of Measurement; TCMS-S, the Spanish version of the Trunk Control Measurement Scale.

## Data Availability

The data associated with the paper are not publicly available but are available from the corresponding author on reasonable request. For additional information please contact javier.lopez3@universidadeuropea.es.

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
