# Peer review of "Trunk Control Measurement Scale (TCMS): Psychometric Properties of Cross-Cultural Adaptation and Validation of the Spanish Version"

_ijerph, 2023, doi:10.3390/ijerph20065144_

Round 1

Reviewer 1 Report

Thank you very much for an opportunity to review the manuscript entitled “Psychometric properties of the Spanish Version of the Trunk 2 Control Measurement Scale (TCMS)”. The current study aimed to develop a Spanish Version of the Trunk Measurement Scale (TCMS-S), to analyze the TCMS-S psychometric properties: validity, reliability, its Standard Error of Measurement (SEM) and Minimal Detectable Change (MDC) in children with cerebral palsy.

Apparently, the manuscript is well-written, however some points of methodology, results and discussion need more information and revision before getting consideration to be published.

Introduction

            Research justification and objectives of the study have been well-written with relevant literature reviews.

Methods and Results

1.     please write clearly about the exclusion criteria of participants with cerebral palsy or their parents/main caregiver as the study excluded 9 patients written on lines 204-208.

2.     The study included children aged 5 to 19 years which is a wide age range. It is interesting to give more information in the results (3.1 participants) how many children were recruited for each age range such as, less than 6 years, 6 – 12 years, more than 12 – 19 years. As trunk control in each age range is different as children continue develop their trunk control from childhood to adolescent.

3.     Lines 98-99, written that ‘The same researcher administers the TCMS-S and the Gross Motor Function Measure-88 (GMFM-88).’  Would this procedure cause any bias from the same researcher perform both tests which were examined their validity? How the authors reduce the possible bias which can happen from the same rater. Please write down this point in the limitation of the study.

4.     Test-retest reliability of this study had been performed using video records for later scoring. This could be one reason why the test-retest reliability was excellent, especially in young children aged less than 6 years. Using video records for repeat scoring has advantages and disadvantages. Authors should aware of good and limited points of using video records and please write more points of view for interpretation of the reliability of test using the video records.

5.     Please give the operational definition of ‘able to sit without support’ used in this study. Description of children with cerebral palsy who are in GMFCS level IV age 4-6, 6-12 and 12-18 years need adaptive seating for trunk control according to Palisano et al 1997. How were trunk control of 8 participants in the GMFCS level IV of this study? How old are they? I have some concerns regarding the result of discriminant validity of the TCMS-S between the GMFCS levels III and IV. The non-significant results could due to small sample size of children with the GMFCS level IV.  Please elaborate the caution when interpret this result.

Discussion

1.     The paragraph on lines 286 regarding the sample size of the study (large) is not relevant to the message in the limitation (“test-retest reliability was studied for a limited sample. A larger sample size would allow for such analyses”).

2.     lines 292-294 needs to make clear discussion with caution regarding children with GMFCS level IV in this study such as how well they control their trunk while sitting without support and limitation of the study regarding the small number of participants. This could be reasons why the previous studies did not recruit participants with GMFCS level IV.

3.     Please rewrite limitation of the study about what have done in the study. What the authors have written are more about suggestion for further studies.

Reviewer 2 Report

Thank you for the opportunity to review the manuscript titled "Psychometric properties of the Spanish Version of the Trunk Control Measurement Scale (TCMS)".

After reviewing it, comment to the authors that the methodology used to carry out the validation of the TCMS in its Spanish version is quite appropriate and it seems that the pscometric properties are excellent.

I would only like to suggest some questions, if the authors consider it so:

- First of all, I would like to suggest a new title that includes the process of this study: Trunk Control Measurement Scale (TCMS): Psychometric Properties of Cross-Cultural Adaptation and Validation of the Spanish Version

- It would be convenient to know how the sample was accessed, just comment that different centers in Madrid are recruited.

- We would like to know for what purpose the Pediatric Disability Inventory Computer Adaptive Test and the CP-QoL are administered, despite the fact that they are scales that measure different parameters from the scale being validated.

- It would be interesting to include an informed consent also for the participants, who, even if they are minors, ethical considerations already include the consent of minors in case they want to participate, as long as their cognitive level allows it.

In general, it is a very well prepared and written study, which should be taken into account for publication since it would resolve, in clinical and research terms, the possibility of measuring trunk control in children with CP and other neurological disorders.

Round 2

Reviewer 1 Report

line 229, a typing error can be corrected (mean age).

Author Response

Dear reviewer, we have corrected the typing error in this latest version that we are sending you. 
Thank you again for your time and dedication. We really believe that the article has been improved thanks to your contributions.
